# Human-Machine Shared Driving Control for Semi-Autonomous Vehicles Using Level of Cooperativeness [note 1]

**DOI:** 10.3390/s21144647

**Published:** 2021-07-07

**Authors:** Anh-Tu Nguyen, Jagat Jyoti Rath, Chen Lv, Thierry-Marie Guerra, Jimmy Lauber

**Affiliations:** 1LAMIH Laboratory UMR CNRS 8201, Université Polytechnique Hauts-de-France, 59300 Valenciennes, France; nguyen.trananhtu@gmail.com (A.-T.N.); guerra@uphf.fr (T.-M.G.); jimmy.lauber@uphf.fr (J.L.); 2Department of Mechanical and Aero-Space Engineering, Institute of Infrastructure Technology Research and Management (IITRAM), Ahmedabad 380026, India; 3School of Mechanical and Aerospace Engineering, Nanyang Technological University, Singapore 639798, Singapore; lyuchen@ntu.edu.sg

**Keywords:** human-machine shared control, polytopic LPV control, lane keeping assistance

## Abstract

This paper proposes a new haptic shared control concept between the human driver and the automation for lane keeping in semi-autonomous vehicles. Based on the principle of human-machine interaction during lane keeping, the level of cooperativeness for completion of driving task is introduced. Using the proposed human-machine cooperative status along with the driver workload, the required level of haptic authority is determined according to the driver’s performance characteristics. Then, a time-varying assistance factor is developed to modulate the assistance torque, which is designed from an integrated driver-in-the-loop vehicle model taking into account the yaw-slip dynamics, the steering dynamics, and the human driver dynamics. To deal with the time-varying nature of both the assistance factor and the vehicle speed involved in the driver-in-the-loop vehicle model, a new $\ell_{\infty }$ linear parameter varying control technique is proposed. The predefined specifications of the driver-vehicle system are guaranteed using Lyapunov stability theory. The proposed haptic shared control method is validated under various driving tests conducted with high-fidelity simulations. Extensive performance evaluations are performed to highlight the effectiveness of the new method in terms of driver-automation conflict management.

## 1. Introduction

Rapid advancements in autonomous vehicle technology have led to the design of several features, such as automated lane keeping [1,2], blind spot monitoring, highway merge, and automated cruise control, among others [3,4,5,6,7]. With the advent of autonomous vehicle technology various areas, such as urban mobility and smart roads [8], collaborative driving and shared driving [9], etc., have been explored. However, dealing with dynamic environments, complex traffic scenarios, weather conditions, connectivity challenges along with legal and ethical issues related to practical implementation of on-road autonomous vehicles still persist. Faced with such challenges, a great deal of research effort on semi-autonomous vehicles, i.e., vehicles with a conditional automation of SAE Level-3, has been performed [10]. The presence of a driver-assistance system (DAS) in semi-autonomous vehicles requires developing control laws that allow the automation to effectively assist the human driver in completing a specified driving task, such as lane keeping, obstacle avoidance, highway merge, etc. However, under unpredictable behaviors and characteristics of the human driver in an open driving environment, the design of effective controllers for DASs of semi-autonomous vehicles is known as a challenging problem [11,12,13]. To deal with this challenge, various control schemes have been proposed under the purview of shared control [14,15,16], i.e., the human driver and the automation cooperates to control the vehicle [17,18,19].

Within the DAS control context, the human-machine interaction (HMI) issue naturally occurs when the human driver and the automation *jointly* performs a driving task [14]. The HMI behavior depends on various characteristics of the human driver. Integrated control for HMI management is achieved by either keeping the driver-in-the-loop (DiL) [13] and by direct steer-by-wire control with driver out of the loop using force control steering [20]. To analyze the influence of such assistance architectures on the human driver, many studies have been conducted with validations on vehicle simulators [13]. Accordingly, the effects of assistive actions on the trust, skill, workload and experience of human drivers have been documented [21,22,23] with analysis of the driver-automation interaction. Note that, in many driving situations, the HMI issue in semi-autonomous vehicles can lead to a conflict between the human driver and the automation, i.e., both the driving agents provide opposing actions to complete the same driving task. These situations arise especially during some extreme maneuvers, such as obstacle avoidance [24], navigating a sharp curve [25], and highway lane change [26], among others. Shared control architectures, considering the HMI management directly in the control design process, have emerged as a promising solution to deal with the driver-automation conflict issue appeared in the driving control process of semi-autonomous vehicles [13,18]. The allocation of the control authority between the automation and the human driver has been proposed in several works, see for instance [14,15,24,27,28,29]. Further research has highlighted that integrating the human characteristics, such as driving skill, style, and workload, in the control loop significantly improves the HMI management and the driving performance [13,23,30].

The authors of Reference [31] have proposed an approach for HMI management based on the level of haptic authority in function of the driver workload and performance [32]. Based on this HMI study, various driver-automation shared control schemes have been developed for shared lane keeping, obstacle avoidance among others [25,33,34,35]. In these works, the conflict issue between the human driver and the automation, which appears in scenarios when their driving objectives are different, can be directly taken into account in the control design. To mitigate the negative impact caused by the driver-automation conflict, the authors of References [21,27] have proposed shared control architectures using the analysis of the intention and the initiative of each driving agent. Based on the cooperative status detection, the smooth transition of the driver-automation control authority between the human driver and the automation was achieved. The authors of Reference [36] have proposed to adapt the control parameters with respect to the individual driver for improving the driving performance of semi-autonomous vehicles. In Reference [28], a haptic control architecture was developed for a smooth transition of the control authority with an adaptation to the driver cognitive workload. It is important to note that the previous works [21,27,28,36] did not consider DiL architectures or include the HMI management in the control loop design.

Motivated by the above control issues, we propose a novel DiL shared driving control architecture for semi-autonomous vehicles. The proposed shared controller is designed in a polytopic linear parameter-varying (LPV) framework [37,38] using a DiL vehicle model. For the development of this latter, the vehicle yaw-slip dynamics are integrated with the lane tracking error dynamics, the steering column dynamics and a dynamic driver model [13]. For HMI management, the cooperative status between the driver and the automation is detected and then used, together with the driver workload, to generate suitably the level of haptic authority required for a given driving situation. Incorporating the information of the level of haptic authority in the control loop, the closed-loop stability with a guarantee on $\ell_{\infty }$-gain performance has been established. The LPV control technique allows handling not only the vehicle speed variations but also the time-varying parameter representing the driver’s need for assistance. To sum up, the contributions of this paper can be summarized as follows.
Using a new concept of level of human-machine cooperativeness, a shared driving control scheme is proposed to manage effectively the conflict issue between the human driver and the automation.For the shared control design, we propose a new Lyapunov-based LPV control method with a reduced conservatism to handle the dynamic control authority factor and the time-varying vehicle speed. Moreover, with a guaranteed $\ell_{\infty }$-gain performance, the proposed shared controller can improve the lane keeping, the vehicle stability, and the human-machine conflict management.

The proposed human-machine shared control method has been validated with a dynamic test track under various road conditions and parametric uncertainties. Extensive evaluations and performance analysis are carried out to demonstrate the effectiveness of the new shared control method in terms of lane tracking, driving comfort, vehicle stability, and also human-machine conflict minimization.

*Notation.* The set of nonnegative integers is denoted by $Z_{+}$. For $N\in Z_{+}$, we denote $I_{N}=\left\{1,\ldots ,N\right\}\subset Z_{+}$. For a matrix *X*, $X^{⊤}$ denotes its transpose, X≻0 means that *X* is positive definite, $\mathrm{He}X=X+X^{⊤}$, and $\lambda_{\min}\left(X\right)$, $\lambda_{\max}\left(X\right)$ denote, respectively, the minimal and maximal eigenvalues of a symmetric matrix *X*. $\mathrm{diag}(X_{1},X_{2})$ denotes a block-diagonal matrix composed of $X_{1}$, $X_{2}$. For a vector $v\in R^{n}$, we denote its 2-norm as $∥v∥=\sqrt{v^{⊤}v}$. For a function $f:R\to R^{n}$, its $\ell_{\infty }$-norm is defined as ${∥f∥}_{\infty }=\sup_{t\in R}∥f\left(t\right)∥$, and $B_{\infty }$ is the set of bounded functions *f*. *I* is the identity matrix of appropriate dimension. The symbol ☆ stands for the terms deduced by symmetry. The time dependency of the variables is omitted when convenient.

## 2. Driver-in-the-Loop Vehicle Modeling

This section presents an integrated DiL vehicle model used for the design of driver-automation shared control. The vehicle and driver parameters are given in Table 1.

### 2.1. Road-Vehicle Dynamics

Under the assumptions of low slip angles and negligible influence of the longitudinal friction forces [3], the front slip angle $\alpha_{f}$ and the rear slip angle $\alpha_{r}$ of the vehicle can be, respectively, expressed by [13]
$$\begin{array}{c} \alpha_{f}=\delta -\beta -\frac{l_{f}\dot{\psi }}{v_{x}},\;\alpha_{r}=\frac{l_{r}\dot{\psi }}{v_{x}}-\beta , \end{array}$$
where $v_{x}$ is the longitudinal speed, β is the lateral side-slip angle, $\dot{\psi }$ is the yaw rate, and δ is the wheel steering angle. Subsequently, the vehicle slip-yaw dynamics based on the well-established bicycle model can be given as follows [3]:(1)$$\begin{array}{c} \begin{array}{cc} \dot{\beta } & =a_{11}\beta +a_{12}\dot{\psi }+a_{15}\delta \\ \ddot{\psi } & =a_{21}\beta +a_{22}\dot{\psi }+a_{25}\delta^{\prime } \end{array} \end{array}$$
with
$$\begin{array}{cccc} a_{11} & =-\frac{C_{f}+C_{r}}{mv_{x}}, & a_{12} & =\frac{l_{r}C_{r}-l_{f}C_{f}}{mv_{x}^{2}}-1,\\ a_{21} & =\frac{l_{r}C_{r}-l_{f}C_{f}}{I_{z}}, & a_{22} & =\frac{l_{r}^{2}C_{r}+l_{f}^{2}C_{f}}{I_{z}v_{x}},\\ a_{15} & =\frac{C_{f}}{mv_{x}}, & a_{25} & =\frac{l_{f}C_{f}}{I_{z}R_{s}}. \end{array}$$

For lane tracking control purposes, the vehicle position error $y_{L}$ and the heading error $\psi_{L}$ at a look-ahead distance $l_{s}$ while traversing a road with a curvature $\rho_{c}$ can be modeled as [25]
(2)$$\begin{array}{c} {\dot{y}}_{L}=\beta v_{x}+l_{s}\dot{\psi }+\psi_{L}v_{x},\;{\dot{\psi }}_{L}=\dot{\psi }-\rho_{c}v_{x}. \end{array}$$

To account for the haptic driver-automation interaction, the following steering column dynamics is also considered [13]:(3)$$\begin{array}{c} {\ddot{\delta }}_{d}=a_{61}\beta +a_{62}\dot{\psi }+a_{65}\delta_{d}-\frac{B_{u}}{I_{s}}{\dot{\delta }}_{d}+\frac{1}{I_{s}}\left(T_{a}+T_{d}\right), \end{array}$$
with $a_{61}=\frac{C_{f}\eta_{t}}{R_{s}^{2}I_{s}}$, $a_{62}=\frac{C_{f}l_{f}\eta_{t}}{R_{s}^{2}v_{x}I_{s}}$, and $a_{65}=-\frac{C_{f}\eta_{t}}{R_{s}^{2}I_{s}}$. For system (Equation 3), $T_{a}$ is the assistance torque, $T_{d}$ is the human driver torque, and $\delta_{d}$ is driver steering angle, i.e., $\delta_{d}=\delta R_{s}$.

### 2.2. Driver Dynamics

For normal driving conditions where the vehicle is negotiating a curve or a straight road section, the two-point visual cues based driver models are generally used to represent the compensatory and anticipatory behaviors. Specifically, these driving behaviors can be, respectively, modeled by the near visual angle $\theta_{n}$ and the far visual angle $\theta_{f}$ as follows [13]:(4)$$\begin{array}{c} \theta_{n}=\frac{y_{L}}{v_{x}T_{p}}+\psi_{L},\;\theta_{f}=\theta_{1}\beta +\theta_{2}\dot{\psi }+\theta_{3}\delta_{d}, \end{array}$$
with $\theta_{1}=\tau_{a}^{2}a_{21}$, $\theta_{2}=\tau_{a}+\tau_{a}^{2}a_{22}$, and $\theta_{3}=\tau_{a}^{2}a_{25}$. The driver anticipation time is defined as $\tau_{a}=\frac{L_{f}}{v_{x}}$, where $L_{f}$ is the far point look-ahead distance. Considering the visual angles $\theta_{n}$ and $\theta_{f}$ defined in (Equation 4) as driver-input for two-level driver models, the following driver model has been proposed and validated in Reference [13]:(5)$$\begin{array}{c} \begin{array}{cc} {\dot{x}}_{d} & =-\frac{1}{t_{i}}x_{d}+\frac{K_{c}\left(t_{l}-t_{i}\right)}{t_{i}}\theta_{n}\\ {\dot{T}}_{d} & =\frac{1}{t_{n}t_{i}}x_{d}-\frac{1}{t_{n}}T_{d}-\frac{K_{c}t_{l}}{t_{i}t_{n}}\theta_{n}+\frac{K_{a}}{t_{n}}\theta_{f} \end{array}, \end{array}$$
where $x_{d}$ is an internal driver state. There also exist other driver models, such as the two-point driver model [25], sensorimotor model [39], cybernetic driver model [40], and far point error model [41], among others which have also been developed. The considered driver model (Equation 5) has been used for validation of shared control works [13] and found to represent the human behaviors accurately.

### 2.3. Integrated Driver-in-the-Loop Vehicle Model

Integrating the vehicle model (Equation 1), the lane positioning dynamics (Equation 2), the steering dynamics (Equation 3) and the driver model (Equation 5), a DiL vehicle model can be obtained as
(6)$$\begin{array}{c} \dot{x}=A\left(v_{x}\right)x+B_{v}u_{v}+E\left(v_{x}\right)w, \end{array}$$
where $x={\left[\begin{array}{cccccccc} \beta & \dot{\psi } & \psi_{L} & y_{L} & \delta_{d} & {\dot{\delta }}_{d} & x_{d} & T_{d} \end{array}\right]}^{⊤}$ is the state vector, $u_{v}=T_{a}$ is the control input, and $w=\rho_{c}$ is the disturbance vector. As in practice, we assume that the disturbance w is unknown but bounded in amplitude, i.e., $w\in B_{\infty }$. The state-space matrices of the system (Equation 6) are given by
$$\begin{array}{cc} A(v_{x}) & =\left[\begin{array}{cccccccc} a_{11} & a_{12} & 0 & 0 & \frac{a_{15}}{R_{s}} & 0 & 0 & 0\\ a_{21} & a_{22} & 0 & 0 & \frac{a_{25}}{R_{s}} & 0 & 0 & 0\\ 0 & 1 & 0 & 0 & 0 & 0 & 0 & 0\\ v_{x} & l_{s} & v_{x} & 0 & 0 & 0 & 0 & 0\\ 0 & 0 & 0 & 0 & 0 & 1 & 0 & 0\\ a_{61} & a_{62} & 0 & 0 & a_{65} & \frac{-B_{u}}{I_{s}} & 0 & \frac{1}{I_{s}}\\ 0 & 0 & a_{73} & a_{74} & 0 & 0 & \frac{-1}{t_{n}} & 0\\ a_{81} & a_{82} & a_{83} & a_{84} & \frac{K_{a}\theta_{3}}{t_{n}} & 0 & \frac{1}{t_{i}t_{n}} & \frac{-1}{t_{n}} \end{array}\right],\\ B_{v} & ={\left[\begin{array}{cccccccc} 0 & 0 & 0 & 0 & 0 & \frac{1}{I_{s}} & 0 & 0 \end{array}\right]}^{⊤},\\ E(v_{x}) & ={\left[\begin{array}{cccccccc} 0 & 0 & -v_{x} & 0 & 0 & 0 & 0 & 0 \end{array}\right]}^{⊤}, \end{array}$$
with
$$\begin{array}{cccc} a_{73} & =\frac{K_{c}\left(t_{l}-t_{i}\right)\left(v_{x}T_{p}-l_{p}\right)}{t_{i}v_{x}T_{p}}, & a_{74} & =\frac{K_{c}\left(t_{l}-t_{i}\right)}{t_{i}v_{x}T_{p}},\\ a_{81} & =\frac{K_{a}\theta_{1}}{t_{n}}, & a_{82} & =-\frac{K_{a}\theta_{2}}{t_{n}},\\ a_{83} & =-\frac{K_{c}t_{l}\left(v_{x}T_{p}-l_{p}\right)}{t_{i}t_{n}v_{x}T_{p}}, & a_{84} & =-\frac{K_{c}t_{l}}{t_{i}t_{n}v_{x}T_{p}}. \end{array}$$

Note that the incorporation of the driver characteristics, including the preview time $T_{p}$ and the anticipation time $\tau_{a}$ allows taking into account the driving style in the driver-automation shared control design.

## 3. Cooperative Framework for Haptic Driver-Automation Interaction

To achieve a better management in terms of human-machine interaction, we propose an cooperativeness indicator to effectively allocate the control authority between the human driver and the automation. To this end, the following index of cooperativeness is defined in a time-window τ as
(7)$$CI\left(\tau \right)=\int_{0}^{\tau }T_{d}\left(t\right)T_{a}\left(t\right)dt,$$
where CI(τ) is the computed cooperative index. Note from (Equation 7) that, when both the driver and the automation have the same driving objective, i.e., perform a similar control action to complete a driving task, then their corresponding torques are generated in the same direction. For example, while driving along a minimal curvature path both the shared controller and the human driver would have the same objective of providing a steering torque to safely negotiate the curve while maintaining the vehicle on the lane. Hence, the cooperative index CI(τ) increases, i.e., a fully cooperative status.

However, when the objective of both driving agents are different, the torques generated by the human driver and the automation are in an opposite direction. These cases generally arise when a sudden maneuver is executed, for instance obstacle avoidance [24], navigating a sharp curve [25], or highway lane change [26]. In these scenarios, the value of CI(τ) decreases, i.e., a non-cooperative status. These driving scenarios should be avoided or reduced to improve the haptic shared driving control performance.

In case of non-cooperative status, the level of haptic authority should be reduced to a minimum level, i.e., the human driver will have a dominant control authority compared to the automation. Note that there also exist situations where the driver and the automation have the same objectives, even the value of CI(τ) is decreasing due to various factors, e.g., sensor drift, noises, and transition between cooperative and non-cooperative status. To avoid false detection of non-cooperative status, the following threshold-based approach is used to categorize the status during shared control process.
*Fully cooperative*: The driver and the automation have same driving objectives, i.e., $CI\left(\tau \right)>\lambda_{c}$.*Non-cooperative*: The human driver and the automation have opposite objectives, which results in a human-machine conflict issue, such as during emergency maneuvers executed by the driver. In such a situation, the cooperative index is also negative, i.e., $CI\left(\tau \right)<\lambda_{c}$. The experimental threshold $\lambda_{c}$ is determined based on shared control evaluations.

The driver need for assistance during a driving task depends on his/her performance characteristics. It has been shown that the required level of haptic authority and the driver performance are *inversely* related [25,31]. Following this HMI study, we introduce the driver activity variable η(τ) taking into account the information of the cooperative index CI(τ) and the measured driver torque $T_{d}$ as
(8)$$\eta \left(\tau \right)=1-e^{-\left(\sigma_{1}CI{\left(\tau \right)}^{\sigma_{2}}T_{dn}^{\sigma_{3}}\right)},$$
where $T_{dn}\in \left[0,1\right]$ is the normalized driver torque and the cooperative index CI(τ)∈[0,1] is also normalized. The parameters $\sigma_{1}$, $\sigma_{2}$, and $\sigma_{3}$, respectively, represent the degree of involvement of the cooperative index CI(τ) and the normalized driver torque $T_{dn}$ in the driver activity variable η(τ). Remark that from (Equation 8) with an increase in the level of cooperativeness CI(τ) or the driver torque $T_{dn}$, the driver activity variable η(τ) increases accordingly to lower the assistance requirement, and vice-versa. A graphical representation of this relationship is depicted in Figure 1.

To analytically replicate this relationship, a dynamic mapping can be defined to compute the assistance factor Γ(η) as
(9)$$\Gamma \left(\eta \right)=\frac{1}{1+|\frac{\eta \left(\tau \right)-p_{3}}{p_{1}}{|}^{2p_{2}}}+\Gamma_{\min}.$$

The time-varying parameter Γ(η) relates the driver performance with the level of haptic authority on the basis of the driver performance for task completion. The parameters $p_{1}=0.355$, $p_{2}=-2$ and $p_{3}=0.5$ are chosen to replicate the U-shaped relationship [31] shown in Figure 1. A minimum assistance level of $\Gamma_{\min}=0.2$ is used to consider the influence of sensor noise, drift, etc. Using the developed mapping in (Equation 9), the assistance torque $T_{a}$ is then modulated as
(10)$$T_{a}=\Gamma \left(\eta \right)u,$$
where the feedback control u is to be designed. From (Equation 6) and (Equation 10), the DiL vehicle model can be rewritten as
(11)$$\begin{array}{c} \dot{x}=A\left(v_{x}\right)x+B\left(\eta \right)u+E\left(v_{x}\right)w, \end{array}$$
with $B\left(\eta \right)={\left[\begin{array}{cccccccc} 0 & 0 & 0 & 0 & 0 & \frac{\Gamma (\eta )}{I_{s}} & 0 & 0 \end{array}\right]}^{⊤}$. The controlled output z of system (Equation 11) is defined to represent both the lane keeping performance and the driving comfort as
(12)$$\begin{array}{c} z={\left[\begin{array}{cccc} a_{y} & \theta_{n} & \theta_{f} & \dot{\delta } \end{array}\right]}^{⊤}. \end{array}$$

For the lane keeping performance, the visual angles $\theta_{n}$ and $\theta_{f}$ given in (Equation 4), respectively, represents the driver’s *compensatory* and *anticipatory* behaviors. The driving comfort is represented by the lateral acceleration $a_{y}≃v_{x}\dot{\psi }$. The steering rate $\dot{\delta }$ is introduced in (Equation 12) to guarantee a desired steering comfort and to improve the vehicle damping response, since all the entries of vector z can be expressed by those of x in (Equation 11) as
(13)$$\begin{array}{c} z=\left[\begin{array}{cccccccc} 0 & v_{x} & 0 & 0 & 0 & 0 & 0 & 0\\ 0 & 0 & 1 & \frac{1}{v_{x}T_{p}} & 0 & 0 & 0 & 0\\ \theta_{1} & \theta_{2} & 0 & 0 & \theta_{3} & 0 & 0 & 0\\ 0 & 0 & 0 & 0 & 0 & \frac{1}{R_{s}} & 0 & 0 \end{array}\right]x. \end{array}$$

Note that the time-varying parameters $v_{x}$ and Γ(η) are directly involved in the dynamics of system (Equation 11) and the performance vector (Equation 13). To achieve an effective human-machine shared control scheme, we propose hereafter an LPV control design guaranteeing some predefined closed-loop specifications.

## 4. LPV Control Design with Guarantee on $\ell_{\infty }$-Gain Performance

This section presents a new LPV control design based on a poly-quadratic Lyapunov function to reduce the design conservatism. Moreover, an $\ell_{\infty }$-gain performance is taken into account in the control design to minimize the disturbance effects. The control result is then applied to the DiL vehicle system (Equation 11).

### 4.1. Control Problem Formulation

For generality, we consider an LPV system of the following state-space realization:(14)$$\begin{array}{cc} \dot{x} & =A(\theta )x+B(\theta )u+E(\theta )w \end{array}$$
$$\begin{array}{cc} z & =C(\theta )x, \end{array}$$
where $x\in R^{n_{x}}$ is the state vector, $u\in R^{n_{u}}$ is the control input, $w\in R^{n_{u}}$ is the disturbance vector, $z\in R^{n_{y}}$ is the controlled output, and $\theta \in R^{p}$ is the measured scheduling variables. It is assumed that the time-varying parameter $\theta ={\left[\begin{array}{c} \theta_{1}\ldots \theta_{p} \end{array}\right]}^{⊤}$ and its rate of variation $\dot{\theta }$ are smooth and, respectively, valued in the following hypercubes:$$\begin{array}{c} \begin{array}{cc} \Omega & =\{{\left(\theta_{1},\ldots ,\theta_{p}\right)}^{⊤}:\;\theta_{j}\in \left[{\underline{\theta }}_{j},{\bar{\theta }}_{j}\right],\;j\in I_{p}\},\\ \Upsilon & =\{{\left({\dot{\theta }}_{1},\ldots ,{\dot{\theta }}_{p}\right)}^{⊤}:\;{\dot{\theta }}_{j}\in \left[{\underline{\upsilon }}_{j},{\bar{\upsilon }}_{j}\right],\;j\in I_{p}\}, \end{array} \end{array}$$
where ${\underline{\theta }}_{j}\leq {\bar{\theta }}_{j}$ (respectively, ${\underline{\upsilon }}_{j}\leq {\bar{\upsilon }}_{j}$) are *known* lower and upper bounds on $\theta_{j}$ (respectively, ${\dot{\theta }}_{j}$), for $j\in I_{p}$. The state-space matrices of system (Equation 14) are continuous on Ω, given by
(15)$$\begin{array}{c} \left[\begin{array}{cc} A(\theta ) & B(\theta )\\ C(\theta ) & E(\theta ) \end{array}\right]=\sum_{i=1}^{N}h_{i}\left(\theta \right)\left[\begin{array}{cc} A_{i} & B_{i}\\ C_{i} & E_{i} \end{array}\right], \end{array}$$
with $N=2^{p}$. The membership functions $h_{i}\left(\theta \right)$, for $i\in I_{N}$, are continuously differentiable and belong to the simplex
$$H=\left\{h\left(\theta \right)\in R^{N}:\;\sum_{i=1}^{N}h_{i}\left(\theta \right)=1,\;h_{i}\left(\theta \right)\geq 0,\;\forall \theta \in \Omega \right\}.$$

Note that, since $(\theta ,\dot{\theta })\in \Omega \times \Upsilon$, one can easily compute the lower bound $\phi_{i1}$ and the upper bound $\phi_{i2}$ of ${\dot{h}}_{i}\left(\theta \right)$ as
(16)$$\begin{array}{c} {\dot{h}}_{i}\left(\theta \right)\in \left[\begin{array}{cc} \phi_{i1}, & \phi_{i2} \end{array}\right],\;\phi_{i1}\leq \phi_{i2},\;i\in I_{N}. \end{array}$$

**Remark** **1.***The sector nonlinearity approach [42] can be used to derive an *exact* polytopic form (Equation 15) for a general LPV system (Equation 14). The membership functions capture the parameter nonlinearities, i.e., they can be a nonlinear function of components of θ(t). Hence, the proposed polytopic LPV method can deal with a larger class of parametric dependencies than, e.g., linear, affine, or rational.*

For control design, we consider an LPV controller as
(17)$$\begin{array}{c} u=K(\theta )x, \end{array}$$
where the gain $K\left(\theta \right)\in R^{n_{u}\times n_{x}}$ is to be designed. From (Equation 14) and (Equation 17), the closed-loop LPV system is rewritten as
(18)$$\begin{array}{c} \dot{x}=\left(A\left(\theta \right)+B\left(\theta \right)K\left(\theta \right)\right)x+E\left(\theta \right)w. \end{array}$$

We are now in the position to formulate the control problem related to the polytopic LPV system (Equation 14).

**Problem** **1.**
*Determine an LPV control law *(Equation 17)* such that the closed-loop system *(Equation 14)* satisfies the following properties.*
*(P1)* *For zero-disturbance system, i.e., w=0, for ∀t≥0, the zero solution of system *(Equation 45)* is *exponentially* stable with a decay rate α>0.**(P2)* 
*The closed-loop system *(Equation 45)* is input-to-state stable with respect to the amplitude-bounded disturbance $w\in B_{\infty }$.*
*(P3)* 
*If w≠0, for ∀t≥0, the state x is uniformly bounded for ∀x(0) and $\forall w\in B_{\infty }$. Moreover, we have*
(19)$$\begin{array}{c} \lim_{t\to \infty }\sup∥z∥\leq {\gamma ∥w∥}_{\infty },\;\;\gamma >0, \end{array}$$
*where the $\ell_{\infty }-$gain γ is specified in Theorem 1. Moreover, if x(0)=0, then $∥z∥\leq {\gamma ∥w∥}_{\infty }$, for ∀t≥0.*



Hereafter, we provide a numerically tractable solution for the above $\ell_{\infty }$-gain control problem.

### 4.2. LPV Control Design with $\ell_{\infty }$-Gain Performance

Using Lyapunov stability theory, the following theorem provides sufficient conditions to design an LPV controller guaranteeing $\ell_{\infty }$-gain performance.

**Theorem** **1.**
*Consider an LPV system *(Equation 14)* with $(\theta ,\dot{\theta })\in \Omega \times \Upsilon$, and a positive scalar α. If there exist symmetric matrices $X\in R^{n_{x}\times n_{x}}$, $Q\in R^{n_{x}\times n_{x}}$, $Q_{i}\in R^{n_{x}\times n_{x}}$, matrices $M_{i}\in R^{n_{u}\times n_{x}}$, for $i\in I_{N}$, and positive scalars ϵ, ν such that the following optimization problem is feasible:*
$$\begin{array}{cc} \; & \underset{\xi_{i}=\left(\nu ,X,Q,Q_{i},M_{i}\right),\;i\in I_{N}}{\mathrm{minimize}}\;\nu \end{array}$$
$$\begin{array}{cc} \; & \mathrm{subject}\;\mathrm{to} \end{array}$$
(20)$$\begin{array}{cc} \; & Q_{i}+Q≻0,\;i\in I_{N}, \end{array}$$
(21)$$\begin{array}{cc} \; & \left[\begin{array}{cc} Q_{i}+Q & \left(Q_{i}+Q\right)C_{j}^{⊤}\\ ⋆ & \nu I \end{array}\right]⪰0,\;i,j\in I_{N}, \end{array}$$
(22)$$\begin{array}{cc} \; & \Phi_{ii}^{kl}≺0,\;\Phi_{ij}^{kl}+\Phi_{ji}^{kl}≺0,\;i<j\in I_{N},k\in I_{N-1},l\in I_{2}.\;\;\; \end{array}$$

*The term $\Phi_{ij}^{kl}$ in (22) is given by*
(23)$$\begin{array}{c} \begin{array}{cc} \Phi_{ij}^{kl} & =\mathrm{He}\left[\begin{array}{ccc} A_{j}\left(Q_{i}+Q\right)+B_{j}M_{i}-\frac{1}{2}\Psi & E_{j} & \epsilon B_{j}M_{i}\\ 0 & -\alpha I & 0\\ Q_{i}+Q-X & 0 & -\epsilon X \end{array}\right]\\ \Psi & =\phi_{kl}\left(Q_{k}+Q-Q_{N}\right)+\frac{1}{N-1}\phi_{Nl}Q-2\alpha \left(Q_{i}+Q\right).\;\; \end{array} \end{array}$$

*Then, controller (Equation 17) with the control gain defined as*
(24)$$\begin{array}{c} K\left(\theta \right)=\sum_{i=1}^{N}h_{i}\left(\theta \right)K_{i},\;K_{i}=M_{i}X^{-1} \end{array}$$
*guarantees that the LPV system (Equation 18) satisfies the closed-loop properties described in Problem 1. Moreover, the guaranteed $\ell_{\infty }$—gain performance is defined as $\gamma =\sqrt{\nu }$.*


**Proof.** For the control design of the LPV system (Equation 18), we consider the parameter-dependent Lyapunov function
(25)$$\begin{array}{c} V\left(x\right)=x^{⊤}Q{\left(\theta \right)}^{-1}x, \end{array}$$
with $Q\left(\theta \right)=\sum_{i=1}^{N}h_{i}\left(\theta \right)\left(Q_{i}+Q\right)$. Condition (Equation 20) guarantees that Q(θ) is positive definite for ∀θ∈Θ. Hence, V(x) is a proper Lyapunov function candidate. Moreover, condition (Equation 22) guarantees that $X+X^{⊤}≻0$, which implies that matrix *X* is nonsingular. This, in turn, guarantees the existence of $X^{-1}$ and, thus, the validity of the control expression (Equation 24). Since $\sum_{i=1}^{N}{\dot{h}}_{i}\left(\theta \right)=0$, for any symmetric matrix *Q*, it follows that
(26)$$\begin{array}{cc} \dot{Q}\left(\theta \right) & =\sum_{k=1}^{N-1}{\dot{h}}_{k}\left(\theta \right)\left(Q_{k}+Q\right)+{\dot{h}}_{N}\left(\theta \right)\left(Q_{N}+Q\right)\\ \; & =\sum_{k=1}^{N-1}{\dot{h}}_{k}\left(\theta \right)\left(Q_{k}+Q-Q_{N}\right)+{\dot{h}}_{N}\left(\theta \right)Q. \end{array}$$For any $\phi_{k1}\leq {\dot{h}}_{k}\left(\theta \right)\leq \phi_{k2}$ in (Equation 16), it follows that
$$\begin{array}{c} {\dot{h}}_{k}\left(\theta \right)=\vartheta_{k1}\left(\theta \right)\phi_{k1}+\vartheta_{k2}\left(\theta \right)\phi_{k2},\;\;k\in I_{N}, \end{array}$$
where
$$\begin{array}{c} \vartheta_{k1}\left(\theta \right)=\frac{\phi_{k2}-{\dot{h}}_{k}\left(\theta \right)}{\phi_{k2}-\phi_{k1}},\;\vartheta_{k2}\left(\theta \right)=\frac{{\dot{h}}_{k}\left(\theta \right)-\phi_{k1}}{\phi_{k2}-\phi_{k1}}. \end{array}$$Note also that $\vartheta_{kl}\left(\theta \right)\geq 0$, $\sum_{l=1}^{2}\vartheta_{kl}\left(\theta \right)=1$, for $\forall k\in I_{N}$. From (Equation 26)–(Equation 28), the term $\dot{Q}\left(\theta \right)$ can be rewritten as
(27)$$\begin{array}{cc} \dot{Q}\left(\theta \right) & =\sum_{k=1}^{N-1}\sum_{l=1}^{2}\left[\vartheta_{kl}\left(\theta \right)\phi_{kl}Q+\frac{1}{N-1}\vartheta_{Nl}\left(\theta \right)\phi_{Nl}Q\right],\;\; \end{array}$$
with $Q=Q_{k}+Q-Q_{N}$. Using expressions (Equation 23) and (Equation 27), condition (Equation 22) implies that
(28)$$\begin{array}{c} \Xi_{ii}\left(\theta \right)≺0,\;\;\Xi_{ij}\left(\theta \right)+\Xi_{ji}\left(\theta \right)≺0,\;\;i,j\in I_{N},\;i<j,\;\;\; \end{array}$$
where
$$\begin{array}{cc} \Xi_{ij}\left(\theta \right) & =\mathrm{He}\left[\begin{array}{ccc} A_{j}Q_{i}+B_{j}M_{i}-\frac{1}{2}\Pi \left(\theta \right) & E_{j} & \epsilon B_{j}\;M_{i}\\ 0 & -\alpha I & 0\\ Q_{i}+Q-X & 0 & -\epsilon X \end{array}\right]\\ \Pi (\theta ) & =\dot{Q}\left(\theta \right)-2A_{j}Q-2\alpha \left(Q_{i}+Q\right). \end{array}$$Since $h_{i}\left(\theta \right)\geq 0$, $\forall i\in I_{N}$, it follows from (Equation 28) that
(29)$$\begin{array}{cc} \sum_{i=1}^{N}h_{i}{\left(\theta \right)}^{2}\Xi_{ii}\left(\theta \right) & +\sum_{i=1}^{N}\sum_{i<j}^{N}h_{i}\left(\theta \right)h_{j}\left(\theta \right)\left(\Xi_{ij}\left(\theta \right)+\Xi_{ji}\left(\theta \right)\right) \end{array}$$
$$\begin{array}{cc} \; & =\sum_{i=1}^{N}\sum_{j=1}^{N}h_{i}\left(\theta \right)h_{j}\left(\theta \right)\Xi_{ij}\left(\theta \right)≺0. \end{array}$$Inequality (Equation 29) can be rewritten in the form
(30)$$\begin{array}{c} \mathrm{He}\left[\begin{array}{ccc} \Sigma_{1}\left(\theta \right)+\alpha Q\left(\theta \right) & E(\theta ) & \epsilon B(\theta )M(\theta )\\ 0 & -\alpha I & 0\\ Q(\theta )-X & 0 & -\epsilon X \end{array}\right]≺0, \end{array}$$
with $\Sigma_{1}\left(\theta \right)=A\left(\theta \right)Q\left(\theta \right)+B\left(\theta \right)M\left(\theta \right)-\frac{1}{2}\dot{Q}\left(\theta \right)$. Multiplying condition (Equation 30) with
$$\left[\begin{array}{ccc} I & 0 & B\left(\theta \right)M\left(\theta \right)X^{-1}\\ 0 & I & 0 \end{array}\right]$$
on the left and its transpose on the right, it follows that
(31)$$\begin{array}{c} \mathrm{He}\left[\begin{array}{cc} \Sigma_{2}\left(\theta \right)+B\left(\theta \right)M\left(\theta \right)X^{-1}Q\left(\theta \right) & E(\theta )\\ 0 & -\alpha I \end{array}\right]≺0, \end{array}$$
with $\Sigma_{2}\left(\theta \right)=A\left(\theta \right)Q\left(\theta \right)+\alpha Q\left(\theta \right)-\frac{1}{2}\dot{Q}\left(\theta \right)$. Pre- and postmultiplying (Equation 31) with$\left[\begin{array}{cc} x^{⊤}Q{\left(\theta \right)}^{-1} & w^{⊤} \end{array}\right]$ and its transpose, we obtain the following condition after some manipulations:
(32)$$\begin{array}{c} \dot{V}\left(x\right)\leq -2\alpha \left(V\left(x\right)-{∥w∥}^{2}\right), \end{array}$$
where $\dot{V}\left(x\right)$ is the time-derivative of the Lyapunov function defined in (Equation 25) along the solution of the closed-loop system (Equation 21). Since $w\in B_{\ell }$, it follows from (Equation 32) that
(33)$$\begin{array}{c} \dot{V}\left(x\right)\leq -2\alpha \left(V\left(x\right)-{∥w∥}_{\infty }^{2}\right). \end{array}$$Multiplying both sides of condition (Equation 33) by $e^{2\alpha t}$, then integrating over $\left[t_{0},t\right]$, it follows that
(34)$$\begin{array}{cc} e^{2\alpha t}V\left(x\left(t\right)\right) & \leq e^{2\alpha t_{0}}V\left(x\left(t_{0}\right)\right)+2\alpha {∥w∥}_{\infty }^{2}\int_{t_{0}}^{t}e^{2\alpha \tau }d\tau \\ \; & =e^{2\alpha t_{0}}V\left(x\left(t_{0}\right)\right)+{∥w∥}_{\infty }^{2}\left(e^{2\alpha t}-e^{2\alpha t_{0}}\right).\;\; \end{array}$$It follows from (Equation 34) that
(35)$$\begin{array}{cc} V(x(t)) & \leq e^{-2\alpha (t-t_{0})}V\left(x\left(t_{0}\right)\right)+{∥w∥}_{\infty }^{2}\left(1-e^{-2\alpha (t-t_{0})}\right)\\ \; & \leq e^{-2\alpha (t-t_{0})}V\left(x\left(t_{0}\right)\right)+{∥w∥}_{\infty }^{2}. \end{array}$$From the definition of the Lyapunov function (Equation 25), we have
(36)$$\begin{array}{c} \varrho_{1}{∥x∥}^{2}\leq V\left(x\right)\leq \varrho_{2}{∥x∥}^{2}, \end{array}$$
with $\varrho_{1}=\min_{\theta \in \Omega }\lambda_{\min}\left(Q{\left(\theta \right)}^{-1}\right)$ and $\varrho_{2}=\max_{\theta \in \Omega }\lambda_{\max}\left(Q{\left(\theta \right)}^{-1}\right)$. Then, it follows from (Equation 35) and (Equation 36) that
$$\begin{array}{c} \varrho_{1}{∥x\left(t\right)∥}^{2}\leq \varrho_{2}e^{-2\alpha (t-t_{0})}∥x\left(t_{0}\right){∥}^{2}+{∥w∥}_{\infty }^{2}, \end{array}$$
which, in turn, implies that
(37)$$\begin{array}{c} ∥x∥\leq \sqrt{\frac{\varrho_{2}}{\varrho_{1}}}e^{-\alpha (t-t_{0})}∥x\left(t_{0}\right)∥+\frac{1}{\sqrt{\varrho_{1}}}{∥w∥}_{\infty }. \end{array}$$Inequality (Equation 37) guarantees that the closed-loop LPV system (Equation 18) is globally bounded for any initial condition x(0) and any $w\in B_{\infty }$. Moreover, if w(t)=0, for $\forall t\in R_{+}$, then system (Equation 18) is *exponentially* stable with a decay rate α. Then, the properties (P1) and (P2) are proved.Multiplying condition (Equation 21) by $h_{i}\left(\theta \right)h_{j}\left(\theta \right)\geq 0$ and summing up for all $i,j\in I_{N}$, we obtain the following condition:
(38)$$\begin{array}{c} \left[\begin{array}{cc} Q(\theta ) & Q\left(\theta \right)C{\left(\theta \right)}^{⊤}\\ ⋆ & \nu I \end{array}\right]⪰0. \end{array}$$Pre- and postmultiplying (Equation 38) with $\mathrm{diag}(Q{\left(\theta \right)}^{-1},I)$ yields
(39)$$\begin{array}{c} \left[\begin{array}{cc} Q{\left(\theta \right)}^{-1} & C{\left(\theta \right)}^{⊤}\\ ⋆ & \nu I \end{array}\right]⪰0. \end{array}$$By Schur complement lemma [43], we show that condition (Equation 39) is equivalent to
(40)$$\begin{array}{c} Q{\left(\theta \right)}^{-1}-\nu^{-1}C{\left(\theta \right)}^{⊤}C\left(\theta \right)⪰0. \end{array}$$Pre- and postmultiplying (Equation 40) with $x^{⊤}$ and its transpose while considering the performance output (Equation 14), we obtain
(41)$$\begin{array}{c} {∥z∥}^{2}\leq \nu V\left(x\right). \end{array}$$It follows from (Equation 35) and (Equation 41) that
(42)$$\begin{array}{c} ∥z\left(t\right)∥\leq \sqrt{\nu V(x\left(t_{0}\right))}e^{-\alpha (t-t_{0})}+\sqrt{\nu }{∥w∥}_{\infty }. \end{array}$$For any initial condition $x(t_{0})$ and any $w\in B_{\infty }$, it follows from (Equation 42) that
(43)$$\begin{array}{c} \lim_{t\to \infty }\sup∥z\left(t\right)∥\leq {\gamma ∥w∥}_{\infty }, \end{array}$$
where the $\ell_{\infty }$-gain in (Equation 19) is defined as $\gamma =\sqrt{\nu }$. Condition (Equation 43) proves the property (P3), which concludes the proof. □

**Remark** **2.***For LPV control design, using the parameter-dependent Lyapunov function (Equation 25) allows to exploit the information of both θ and $\dot{\theta }$, represented by the bounds $\phi_{kl}$, for $k\in I_{N}$, $l\in I_{2}$, to reduce the design conservatism. Indeed, if Q=0, $Q_{1}=\cdots =Q_{N}=P$, then we directly recover from (Equation 25) the classical quadratic Lyapunov function $V\left(x\right)=x^{⊤}Px$. Moreover, if (Equation 22) is feasible for *arbitrarily* large values of $|\phi_{kl}|$, then the only possible solution is such that $Q_{1}\approx \cdots \approx Q_{N}$ and Q≈0 to minimize the effect of the term $\phi_{kl}\left(Q_{k}+Q-Q_{N}\right)+\frac{1}{N-1}\phi_{Nl}Q$ in (Equation 23). Hence, the proposed results include those derived from quadratic or poly-quadratic Lyapunov functions $V\left(x\right)=x^{⊤}\sum_{i=1}^{N}h_{i}\left(\theta \right)P_{i}x$. Similar remarks on the design conservatism when using parameter-dependent Lyapunov functions can be found in Reference [44].*

**Remark** **3.**
*The control design in Theorem 1 is reformulated as an optimization problem under LMI-based constraints (Equation 20)–(Equation 22), which can be solved with standard solvers [43].*


### 4.3. Application to Human-Automation Shared Driving Control

For LPV control design, we first represent the DiL vehicle model (Equation 11) in a polytopic LPV form. There are four time-varying parameters involved in the dynamics of system (Equation 11): $v_{x}$, $\frac{1}{v_{x}}$, $\frac{1}{v_{x}^{2}}$ and Γ(η). Note that the number of vertices of a polytopic LPV model increases exponentially according to the number of time-varying parameters. Indeed, if these four parameters are *independently* considered as scheduling parameters, then we obtain a polytopic LPV model with $2^{4}=16$ vertices. To reduce the numerical complexity and also the design conservatism, the relationship between $v_{x}$, $\frac{1}{v_{x}}$ and $\frac{1}{v_{x}^{2}}$, with $v_{x}\in \left[v_{\min},v_{\max}\right]$, should be exploited. To this end, we introduce the new time-varying parameter ζ and then using Taylor approximation to represent $v_{x}$, $\frac{1}{v_{x}}$ and $\frac{1}{v_{x}^{2}}$ as follows [5]:(44)$$\begin{array}{c} \begin{array}{cc} \frac{1}{v_{x}} & =\frac{1}{v_{0}}+\frac{1}{v_{1}}\zeta ,\;v_{x}≃v_{0}\left(1-\frac{v_{0}}{v_{1}}\zeta \right),\\ \frac{1}{v_{x}^{2}} & ≃\frac{1}{v_{0}^{2}}\left(1+2\frac{v_{0}}{v_{1}}\zeta \right), \end{array} \end{array}$$
with
$$v_{0}=\frac{2v_{\min}v_{\max}}{v_{\max}+v_{\min}},\;v_{1}=\frac{-2v_{\min}v_{\max}}{v_{\max}-v_{\min}}.$$

Remark that ζ=−1 if $v_{x}=v_{\min}=5$ [m/s] and ζ=1 if $v_{x}=v_{\max}=25$ [m/s]. Substituting expressions in (Equation 44) into system (Equation 11), we obtain a DiL vehicle model with the scheduling vector as $\theta ={\left[\begin{array}{cc} \zeta & \Gamma (\eta ) \end{array}\right]}^{⊤}\in R^{2}$. Then, the corresponding polytopic LPV has only $2^{2}=4$ vertices, defined as
(45)$$\begin{array}{c} \begin{array}{cc} \dot{x} & =\sum_{i=1}^{4}h_{i}\left(\theta \right)\left(A_{i}x+B_{i}u+E_{i}w\right)\\ z & =\sum_{i=1}^{4}h_{i}\left(\theta \right)C_{i}x \end{array}, \end{array}$$
where the local matrices $\left(A_{i},B_{i},C_{i},E_{i}\right)$, and the membership functions $h_{i}\left(\theta \right)$, for $i\in I_{4}$, can be directly obtained from the sector nonlinearity approach [42], which are omitted here for brevity. To limit the kinematic acceleration, the following bounds of the vehicle acceleration are considered [44,45]:(46)$$a_{\min}\leq a_{x}={\dot{v}}_{x}\leq a_{\max},\;a_{\max}=-a_{\min}=4\;[m/s{}^{2}].$$

Then, it follows from (Equation 44) and (Equation 46) that
(47)$$\frac{a_{\min}}{a_{0}}\leq \dot{\zeta }\leq \frac{a_{\max}}{a_{0}},\;a_{0}=-\frac{v_{0}^{2}}{v_{1}}.$$

Moreover, from the analytical expression of the assistance factor Γ(η) in (Equation 9), we can derive the following bounds:(48)$$\gamma_{0\min}\leq \dot{\Gamma }\left(\eta \right)\leq \gamma_{0\max},$$
with $\gamma_{0\max}=-\gamma_{0\min}=6$. As discussed in Remark 2, the bounds (Equation 47)–(Equation 48) allows using a parameter-dependent Lyapunov function for LPV control design to reduce the conservatism of the control results.

## 5. Validations and Performance Analysis

This section presents comprehensive evaluations and performance analysis of the proposed shared lane keeping assistance controller. The validations have been performed on a multi-degrees of freedom nonlinear vehicle simulator with nonlinear Brush tire friction forces [46] developed and implemented in MATLAB-SIMULINK platform.

### 5.1. Validation Setup and Performance Criteria

The performance of the proposed human-machine shared controller has been evaluated for lane keeping under different road friction conditions and parametric uncertainties. The simulated dynamic test track is with various varying curvatures as depicted in Figure 2a. To simulate the behaviors of the human driver, the two-point driver model in Reference [25] has been employed in the simulations. The driver torque $T_{d}^{v}$ issued from this model, i.e., the virtual driver toque used to represent a human driver, can be given as a linear combination of the driver’s anticipatory and compensatory actions for a specified look-ahead distance. By varying the anticipatory and compensatory gains, i.e., $K_{a}^{v}$ and $K_{c}^{v}$, respectively, the characteristics of various drivers can be replicated. Note that the response of the two-point driver model [25] is not exactly similar to the driver model (Equation 5) used for the shared control design. For illustrations, we present the comparison of torques generated by both the driver models with the same anticipatory and compensatory gains in Figure 2b.

To evaluate the lane keeping performance of the proposed shared controller, we compute the maximum and root-mean-square (RMS) values of the tracking errors $y_{L}$ and $\psi_{L}$. For the driving comfort, the indicator on the steering rate is used. Concerning the vehicle stability analysis, the maximum and RMS values of the vehicle yaw rate are computed. For the driver-automation shared control performance, similar to Reference [26], the following indicators are defined for a time interval of τ:(49)$$\begin{array}{c} \mathrm{SC}=\frac{\int_{0}^{\tau }y_{L}\left(t\right)dt}{T_{d_{pow}}},\;\;\mathrm{SW}=\frac{1}{\tau }\int_{0}^{\tau }T_{a}\left(t\right)T_{d}\left(t\right){\dot{\delta }}_{d}\left(t\right)dt,\;\; \end{array}$$
where the steering power of the human driver is given by $T_{d_{pow}}=\frac{1}{\tau }\int_{0}^{\tau }T_{d}{\left(t\right)}^{2}dt$. Note that SC represents the steering comfort satisfaction levels of the driver while SW represents the steering workload. For a high value of SC, the effort generated by the human driver results in a good steer-ability and, thus, a high driving satisfaction. The steering workload SW is representative of the effort generated by both agents simultaneously for completing the driving task. Typically, higher values of negative SW indicate a poor assistance provided to the human driver [13]. Moreover, the following performance indicators are also considered:(50)$$\begin{array}{c} \mathrm{PRatio}=\frac{T_{d_{pow}}}{T_{a_{pow}}},\;\;\;\mathrm{Conflict}=T_{a}T_{d}, \end{array}$$
where PRatio represents the efforts generated by both agents, $T_{a_{pow}}=T_{a_{pow}}=\frac{1}{\tau }\int_{0}^{\tau }T_{a}{\left(t\right)}^{2}dt$ is the power of the assistance system, and the torque product Conflict indicates the human-machine conflict. Note that, when the values of PRatio >1, the assistance provided by the automation is less than that of the driver, and inversely for PRatio <1. Moreover, the driver-automation conflict is present when Conflict<0.

### 5.2. Shared Control Performance Evaluation

For illustrations, the performance analysis of the shared driving control performed on the road curvature shown in Figure 2a with a surface friction coefficient of 1 is presented. The controlled lateral acceleration of the vehicle during this maneuver is depicted in Figure 2c, which indicates the safe handling limits. Under such operating conditions, the controlled states of the vehicle are shown in Figure 3.

We can see that the controlled states are constrained within a safe vehicle operating condition. The lane keeping performance is also guaranteed by the low magnitude of the tracking errors $y_{L}$ and $\psi_{L}$. The maximum and RMS values of these errors are, respectively, given by $|y_{L}{|}_{\max}=0.522$ [m], $|\psi_{L}{|}_{\max}=0.063$ [rad], and $|y_{L}{|}_{RMS}=0.338$ [m], $|\psi_{L}{|}_{RMS}=0.024$ [rad]. These results confirm that the controlled vehicle is maintained around the lane center. Similarly, the maximum and RMS values of the steering rate are, respectively, obtained as $|{\dot{\delta }}_{d}{|}_{\max}$ = 1.686 [rad/s] and $|{\dot{\delta }}_{d}{|}_{RMS}=0.407$ [rad/s], which shows a good driver comfort level while completing the driving task. The vehicle stability is also guaranteed with small computed indexes for the yaw rate are ${\left|\psi \right|}_{\max}$ = 0.2597 [rad/s] and ${\left|\psi \right|}_{RMS}$ = 0.1641 [rad/s]. Observe that even during sharp curves of radius 25 [m], the maximum values of the yaw rate and the steer-rate do not increase beyond their respective maximal levels ${\dot{\psi }}_{\max}=0.55$ [rad/s] and ${\dot{\delta }}_{f_{\max}}=0.15$ [rad/s], which also indicates a good control performance.

The above lane tracking, driver comfort, and vehicle stability performance is obtained with the driver and assistance torques presented in Figure 4.

The magnitude of the internal driver state $x_{d}$ shows that the steering wheel correction performed by the driver based on his/her perception of the road conditions is low, thus ensuring enhanced driver comfort. Similarly, the illustrations of the assistance and driver torques presented in Figure 4b show that the assistance torque generally has higher magnitude than the driver torque. The monitored driver activities and the corresponding sharing of authority allocation factor are presented in Figure 5a,b, respectively. The product of the assistance and driver torques, considered as an indicator of the conflict between two driving agents, is also shown in Figure 5c.

We can see that, when the conflict is present, i.e., Conflict<0, only a low level of haptic authority is provided to the human driver, and he/she completely takes over the vehicle control. In other scenarios, the assistance torque is modulated by the driver physical workload. This reduces the driver-automation conflict, as shown in Figure 5c. To evaluate the quality of the shared control, the computed values of the metrics presented in (Equation 49) and (Equation 50) are obtained as PRatio = 0.0386, SC = 0.0848 [N${}^{-2}$m${}^{-1}$], and SW = −1.395 [N${}^{2}$m${}^{2}$rad/s] over the whole driving maneuver. Further, the minimum value of the conflict was obtained as Conflict${}_{\min}$ = −2.9392 [N${}^{2}$m${}^{2}$], which is greater than the design threshold $\lambda_{c}=-3$ [N${}^{2}$m${}^{2}$]. These results highlight a good quality of shared control and conflict minimization between both driving agents.

### 5.3. Control Robustness w.r.t. Modeling Uncertainty

There exists a modeling mismatch between the DiL vehicle model (Equation 6) used for shared control and the DiL vehicle model used for simulations. To evaluate the control robustness with respect to the modeling uncertainty, the performance metrics on lane tracking, vehicle stability, driver comfort, and sharing of authority corresponding to varying road friction conditions and to the presence of uncertainty in *m*, $I_{z}$, $I_{s}$, are computed for the test track depicted in Figure 2a and presented in Table 2. To consider the road friction conditions in the validation tests, the front and rear tire-road forces are, respectively, modeled as $F_{yf}=\mu C_{f}\alpha_{f}$ and $F_{yr}=\mu C_{r}\alpha_{r}$, where μ is the road friction. Similarly, to account for varying driver behaviors, the results for various performance metrics considering uncertainty in the driver parameters $K_{a}$ and $K_{c}$ for the human driver, i.e., two-point driver model, are presented in Table 3. Note also that these parametric uncertainties are only considered for the test scenarios and not taken into account in the control design.

For comparisons with the proposed shared controller (CITDN), the results obtained with the following control schemes are also presented:
Auto: Autonomous controller with no driver, i.e., $T_{d}=0$.Auto-FA: Autonomous controller with driver present and full assist always provided.HMI-FA: Shared DiL controller with full assist always provided, i.e., Γ(η)=1.

Across different road conditions and uncertainties, the RMS values of different metrics exhibit negligible variance for all considered controllers. However, the maximum values of these metrics, which help in the performance analysis for extreme conditions, exhibit a significant difference for all controllers, as shown in Table 2. Similar conclusions about the performance of all controllers can be drawn from the presented results in Table 3 concerning the driver behaviors. For the high friction road condition, i.e., μ=1, even with 25% variations in the parameters $K_{a}$ and $K_{c}$ the lane keeping metrics and the HMI metrics indicate a good performance across all considered controllers. Especially, with a decreasing road friction condition, the instantaneous human-machine conflict represented by the minimum value of the cooperative index decreases sharply for all the control architectures. Such performance across the presented driver uncertainties, thus, accounts for the variations of driver behaviors which can be mapped based on the gains $K_{a}$ and $K_{c}$ as previously discussed. From the presented results, it can be deduced that for the considered uncertainty scenarios, the proposed CITDN controller outperforms the other controllers.

Considering a dry road condition, i.e., μ=1, with low parametric uncertainties, the Auto-FA controller offers the best lane tracking performance. However, this controller poorly fares in achieving high driver comfort, vehicle stability, and quality of shared control. In contrast, the proposed CITDN controller outperforms other controllers in all aspects. However, the instantaneous conflict minimization by the CITDN controller is also affected for slippery road conditions with parametric uncertainties. Thus, with a decrease in the value of road friction coefficient, the value of $CI_{\min}$ crosses the predefined threshold $\lambda_{c}=-3$.

## 6. Conclusions and Future Works

A new linear parameter varying design for shared driving control with adaptation to level of cooperativeness and driver workload has been proposed for semi-autonomous vehicles. To take into account the driver characteristics in the control design, a dynamic driver model is considered to construct a driver-in-the-loop vehicle model. The haptic shared control strategy is proposed based on a new index of cooperativeness and the driver need for assistance with respect to his/her driving activity. Using polytopic linear parameter varying control technique, together with Lyapunov stability arguments, the proposed shared controller is able to deal with the time-varying vehicle speed and a dynamic modulation factor used to manage the driver-automation conflict issue. The new shared controller provides a good performance with small lane tracking errors, enhanced driver comfort, and good sharing of control authority over a dynamic test track with various parametric uncertainties. Extensive comparisons with other shared control architectures and fully autonomous controllers show that the proposed shared control scheme leads to the best performance across all considered evaluation metrics. For future works, the validation of the proposed shared control architecture on a driving simulator and testing for extreme maneuvers, such as obstacle avoidance and highway merge, will be explored. For real-time validations, dealing with the estimation of vehicle variables for feedback control design, e.g., using LPV observers, and the control robustness with respect to modeling uncertainties will be of crucial importance, which requires further investigations. 

## Figures and Tables

**Figure 1 sensors-21-04647-f001:**
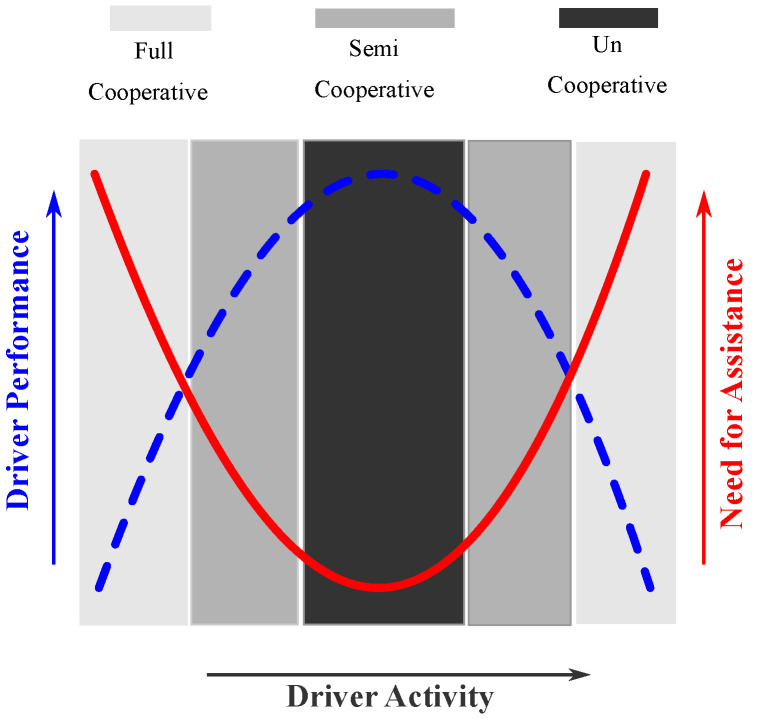
Representation of the driver activity, the driver performance, and the required level of haptic authority [31].

**Figure 2 sensors-21-04647-f002:**
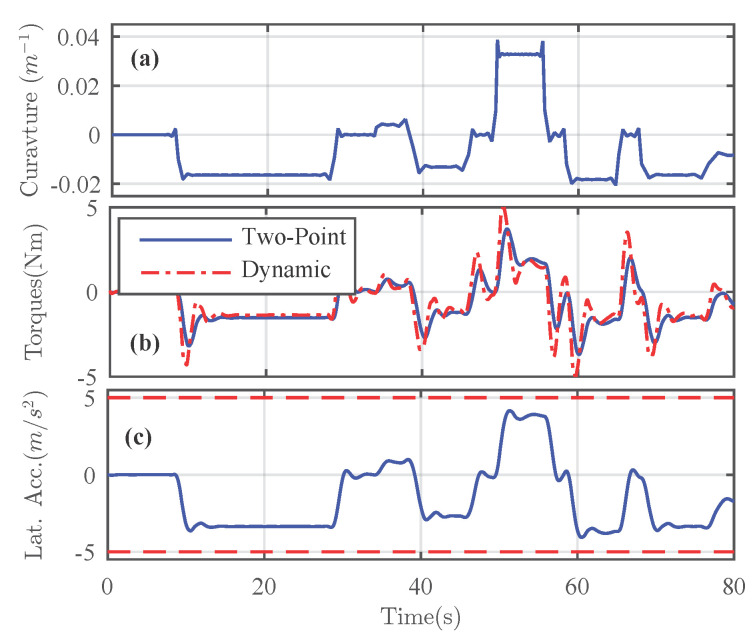
(**a**) Test-track with varying curvatures. (**b**) Driver model comparison: two-point driver model used to replicate the human driver and dynamic driver model used for shared control design. (**c**) Lateral acceleration along the dynamic test.

**Figure 3 sensors-21-04647-f003:**
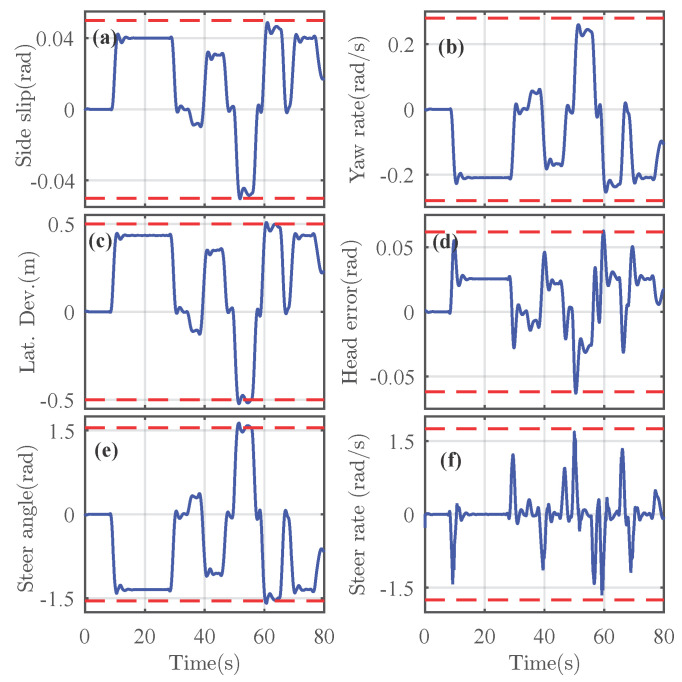
Controlled vehicle states. (**a**) Sideslip angle β. (**b**) Yaw rate $\dot{\psi }$. (**c**) Lateral deviation $y_{L}$. (**d**) Heading angle $\psi_{L}$. (**e**) Steering angle $\delta_{d}$. (**f**) Steering rate ${\dot{\delta }}_{d}$.

**Figure 4 sensors-21-04647-f004:**
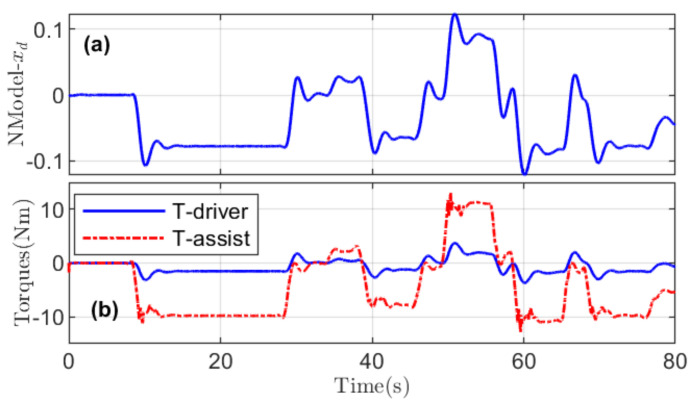
(**a**) Internal driver state $x_{d}$ for the driver model. (**b**) Driver and assistance torques generated for completing the driving task.

**Figure 5 sensors-21-04647-f005:**
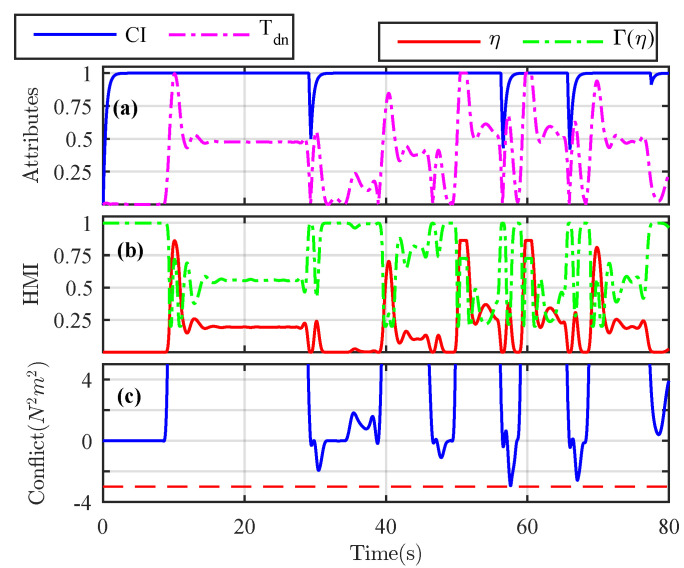
(**a**) Cooperative index and normalized driver torque. (**b**) Driver performance and provided level of haptic authority. (**c**) Conflict between the human driver and the automation represented by Conflict $=T_{a}T_{d}$.

**Table 1 sensors-21-04647-t001:** Vehicle and driver model parameters.

Symbol	Description	Value
*m*	total mass of the vehicle	2025 kg
$l_{f}$	distance from CoG to front axle	1.3 m
$l_{r}$	distance from CoG to rear axle	1.6 m
$l_{s}$	look-ahead distance	5 m
$\eta_{t}$	tire length contact	0.052 m
$I_{z}$	vehicle yaw moment of inertia	2800 kgm${}^{2}$
$I_{s}$	steering moment of inertia	0.05 kgm${}^{2}$
$R_{s}$	steering gear ratio	17.3
$B_{s}$	steering system damping	2.5 N/rad
$C_{f}$	front cornering stiffness	42,500 N/rad
$C_{r}$	rear cornering stiffness	57,000 N/rad
$T_{p}$	driver preview time	1.2 s
$t_{i}$	compensatory lead time	0.31 s
$t_{l}$	compensatory lag time	1.35 s
$t_{n}$	lag time	0.14 s
$K_{a}$	driver anticipatory parameter	5.15
$K_{c}$	driver compensatory parameter	1.96

**Table 2 sensors-21-04647-t002:** Control robustness w.r.t. vehicle parametric uncertainties on *m*, $I_{z}$, and $I_{s}$.

Road	Vehicle Uncertainties	Controller	$|y_{L}|{}_{\max}$ [m]	$|\psi_{L}|{}_{\max}$ [rad]	$|{\dot{\delta }}_{d}\left|{}_{\max}\right|$ [rad/s]	$|\dot{\psi }|{}_{\max}$ [rad/s]	PRatio [–]	SC [N${}^{-2}$m${}^{-1}$]	SW [N${}^{2}$m${}^{2}$rad/s]	${\mathrm{CI}}_{\min}$ [N${}^{2}$m${}^{2}$]
μ=1	5%	Auto	0.545	0.075	2.024	0.286	–	–	–	–
		Auto-FA	0.499	0.071	1.971	0.278	0.041	0.071	−2.356	−9.335
		HMI-FA	0.536	0.065	1.593	0.263	0.039	0.087	−1.549	−3.371
		CITDN	0.510	0.063	1.555	0.259	0.039	0.083	−1.378	−3.351
	25%	Auto	0.540	0.074	2.029	0.283	–	–	–	–
		Auto-FA	0.487	0.069	2.015	0.272	0.042	0.074	−2.201	−8.252
		HMI-FA	0.517	0.064	1.865	0.256	0.039	0.089	−2.123	−2.891
		CITDN	0.509	0.063	1.585	0.257	0.037	0.087	−1.324	−2.944
μ=0.5	5%	Auto	0.6287	0.0792	2.3133	0.3018	–	–	–	–
		Auto-FA	0.578	0.076	2.625	0.297	0.056	0.054	−3.247	−23.558
		HMI-FA	0.668	0.072	2.265	0.282	0.055	0.066	−2.335	−15.474
		CITDN	0.633	0.066	2.123	0.278	0.053	0.066	−2.198	−14.575
	25%	Auto	0.6247	0.0785	2.3901	0.3002	–	–	–	–
		Auto-FA	0.553	0.073	2.813	0.287	0.054	0.059	−3.061	−23.041
		HMI-FA	0.612	0.066	2.220	0.273	0.049	0.077	−2.612	−8.214
		CITDN	0.629	0.067	2.169	0.276	0.049	0.071	−2.095	−13.032

**Table 3 sensors-21-04647-t003:** Control robustness w.r.t. driver uncertainties on $K_{a}$ and $K_{c}$.

Road	Vehicle Uncertainties	Controller	$|y_{L}|{}_{\max}$ [m]	$|\psi_{L}|{}_{\max}$ [rad]	$|{\dot{\delta }}_{d}\left|{}_{\max}\right|$ [rad/s]	$|\dot{\psi }|{}_{\max}$ [rad/s]	PRatio [–]	SC [N${}^{-2}$m${}^{-1}$]	SW [N${}^{2}$m${}^{2}$rad/s]	${\mathrm{CI}}_{\min}$ [N${}^{2}$m${}^{2}$]
μ=1	5%	Auto-FA	0.503	0.074	2.049	0.278	0.041	0.071	−2.357	−10.143
		HMI-FA	0.539	0.065	1.592	0.263	0.039	0.086	−1.458	−3.492
		CITDN	0.514	0.063	1.545	0.259	0.039	0.083	−1.379	−3.336
	25%	Auto-FA	0.503	0.071	2.056	0.278	0.041	0.071	−2.364	−10.505
		HMI-FA	0.538	0.065	1.593	0.263	0.039	0.086	−1.459	−3.662
		CITDN	0.511	0.063	1.551	0.259	0.039	0.083	−1.381	−3.501
μ=0.5	5%	Auto-FA	0.583	0.076	2.589	0.296	0.056	0.053	−3.522	−23.388
		HMI-FA	0.673	0.075	2.307	0.283	0.056	0.067	−2.324	−16.577
		CITDN	0.633	0.066	2.108	0.0278	0.053	0.066	−2.197	−14.545
	25%	Auto-FA	0.583	0.076	2.581	0.295	0.056	0.053	−3.523	−22.195
		HMI-FA	0.673	0.075	2.311	0.283	0.056	0.066	−2.321	−17.376
		CITDN	0.633	0.066	2.115	0.278	0.054	0.066	−2.195	−15.133

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
