# Peer review of "Human-Machine Shared Driving Control for Semi-Autonomous Vehicles Using Level of Cooperativeness â€"

_sensors, 2021, doi:10.3390/s21144647_

Round 1
Reviewer 1 Report
The paper proposes a method for cooperative driving (between human and AI) that assesses the required degree of haptic authority according the the control characteristics of the human driver. I recommend the following amendments to be implemented prior to publication.
- Do not mention MATLAB/SIMULINK in the abstract. This is not necessary since it is not a major highlight of the paper, and the simulations should be independent of the computing framework employed.
- It is better not to use acronyms in the abstract and conclusions.
- Please note that besides algorithmic approaches, alternate steering control systems have been proposed to better integrate the human and the driver: ' communication and interaction with semiautonomous ground vehicles by force control steering'. This should be discussed.
- 'Level of assistance': there is already settled nomenclature to refer to this concept (loose rein/tight rein or level of haptic authority).
- 'DiL' not defined.
- Why the proposed driver model was chosen? The authors should give more details of how this model is representative and reference other models, such as 'A new model of human steering using far-point error perception and multiplicative control'. Also There should be some discussion as to how the proposed approach is dependent of the driver model used.
- My major concern is that the paper is based on simulation and not on driver simulator experiments. Thus the applicability of the work needs to be further justified as to explain why that was not necessary.
Reviewer 2 Report
the paper written by the authors is structured in a clear and organized way. I recommend increasing the introductory part, referring to self-driving vehicles as everything can improve in the near future. I recommend reviewing the format of the references. to increase the bibliography especially by referring to future works. check the part of the conclusions, I suggest you describe them in a more extensive way.
the english language is quite clear and well written. check some typos in the text.
I recommend the following works to be included in the bibliography
Smart roads: An overview of what future mobility will look like
Trubia, S., Severino, A., Curto, S., Arena, F., Pau, G.
, , , ,
Decision tree method to analyze the performance of lane support systems
Pappalardo, G., Cafiso, S., Di Graziano, , ,
Reviewer 3 Report
The paper presents a polytopic LPV model based control design for semi autonomous vehicles. The presented results are interesting and are in line with the scope of the journal. My specific comments are the following:
- Please define DiL.
- The reference in line 192 is missing.
- Is it reasonable to assume that all states are measurable or state estimation is needed as well?
- Can the proposed method be extended to grid-based LPV representation as well? What are the benefits of using polytopic approach instead of grid-based representation?
- The polytopic model is obtained by the sector nonlinearity approach. How does such derivation influence the final performance of the controller? For example a polytopic model can be derived via Tensor Product transformation (Yam, Y.: Fuzzy Approximation via Grid Point Sampling and Singular Value Decomposition, IEEE Transactions on Systems, Man, and Cybernetics, 27:(6), pp. 933-951 (1997); Baranyi, P.: TP model transformation as a way to LMI-based controller design, IEEE Transactions in Industrial Electronics, 51:(2), pp. 387-400 (2004); Baranyi, P., Takarics, B.: Aeroelastic Wing Section Control via Relaxed Tensor Product Model Transformation Framework, Journal of Guidance, Control, and Dynamics, 37:(5), pp. 1671-1678, (2014); Takarics, B., Baranyi, P.: Tensor Product Model Based Control of a Three Degrees-of-Freedom Aeroelastic Model. Journal of Guidance, Control, and Dynamics, 36:(5), pp. 1527-1533. (2013)) has been successfully applied to derive polytopic models of nonlinear systems. It has been also shown that the polytopic representation significantly influences the feasibility of the LMIs.
- Is there any sensor noise in the validation example? What about computational time delays?
- What is the reason for not including the parametric uncertainties in the control design? Polytopic LPV systems work well to construct polytopic uncertain models.
I recommend revising the paper based on the comments above.
Round 2
Reviewer 1 Report
The new version of the paper is much improved and I recommend it for publication.
Reviewer 3 Report
The authors answered all my comments of the initial version of the paper in detail. Therefore, I recommend accepting the paper in the current form.